# Impact of Land Use/Cover Changes on Soil Erosion by Wind and Water from 2000 to 2018 in the Qaidam Basin

Xue Cao [1,2,3,4,5], Yuzhuo Cheng [1,2,3], Juying Jiao [1,2,3,6,*], Jinshi Jian [6], Leichao Bai [4,5,6], Jianjun Li [6] and Xiaowu Ma [6]

1    The Research Center of Soil and Water Conservation and Ecological Environment, Chinese Academy of Sciences, Ministry of Education, Xianyang 712100, China; caoxue19@mails.ucas.ac.cn (X.C.); cyz9711@nwafu.edu.cn (Y.C.)
2    Institute of Soil and Water Conservation, Chinese Academy of Sciences and Ministry of Water Resources, Xianyang 712100, China
3    University of Chinese Academy of Sciences, Beijing 100049, China
4    School of Geographical Sciences, China West Normal University, Nanchong 637009, China; beleit@cwnu.edu.cn
5    Sichuan Provincial Engineering Laboratory of Monitoring and Control for Soil Erosion in Dry Valleys, China West Normal University, Nanchong 637009, China
6    Institute of Soil and Water Conservation, Northwest A&F University, Xianyang 712100, China; jinshi@vt.edu (J.J.); lijianjun@nwafu.edu.cn (J.L.); maxiaowu@nwafu.edu.cn (X.M.)
*    Correspondence: jyjiao@ms.iswc.ac.cn

**Abstract:** Assessing the impact of land use and land cover change (LUCC) on soil erosion by wind and water is crucial for improving regional ecosystem services and sustainable development. In his study, the Revised Wind Erosion Equation (RWEQ) and Revised Universal Soil Loss Equation (RUSLE) were used to reveal changes in the extent of soil erosion by wind and water in the Qaidam Basin from 2000 to 2018 and the impact of LUCC on them. From 2000 to 2018, with global climate change, the areas and intensities of soil erosion by wind decreased, whereas those of soil erosion by water increased. With increased human activities, approximately 12.96% of the total area underwent conversion of the type of use: the areas of cropland, woodland, grassland, and construction land increased, whereas the areas of shrubbery, desert, and other unused land decreased. Land use/cover changes are positive to the soil erosion of water but negative to the soil erosion of wind. Among them, the changes in vegetation coverage of other unused land and grassland contributed to 83.19% of the total reduction in soil erosion by water. Converting other unused land to grassland reduced the total reductions in soil erosion by wind by 94.69%. These results indicate that the increase in vegetative cover and area of grasslands in the Qaidam Basin had a positive impact on the reduction in soil erosion. It is recommended that the arrangement of grasses, shrubs, and trees be optimized to prevent compound erosion by wind and water for protecting regional ecological environments.

**Keywords:** Qaidam Basin; soil erosion; RWEQ; RUSLE; land use and land cover change

## 1. Introduction

Soil erosion seriously threatens the stability of ecosystems and sustainable development of the social economy [1,2]. Erosion by wind and erosion by water are two common types of soil erosion. Globally, 70% of countries and regions in the world are affected by desertification disasters and soil erosion [3]. Soil erosion by wind leads to massive losses of clay, silt, and soil nutrients in arid and semi-arid areas. It sometimes even causes sand and dust storms, which adversely influence a range of ecosystem parameters [4,5]. Soil erosion by water causes the stripping, migration, and deposition of soil minerals and organic matter, resulting in reduced soil fertility and environmental pollution [6]. The combined soil erosion by wind and water at temporal and spatial scales often leads to

complex compound soil erosion [7–9]. However, a comprehensive analysis of compound erosion on the Qinghai–Tibet Plateau is still lacking.

Global climate change and increased human activity pose challenges for soil erosion research [10,11]. Different climatic factors affect soil erosion, mainly through their own variations [12,13]. Improvements in rainfall, temperature, and other climatic conditions promote plant growth and increase vegetation coverage, thereby indirectly improving soil resistance to erosion [14]. In addition to climatic conditions, land use and land cover change (LUCC) represents the macroscopic manifestation of human activities [15]. The changes in land use and vegetation caused by humans affect soil erosion by wind and water [5]. From a global perspective, soil and water conservation measures often play an important role in protecting ecosystems [16]. It has been shown that planting trees and grasses effectively reduces runoff, intercepts sand, and decreases soil erosion [17]. Human activities, such as reclamation, overgrazing, and logging, which are usually beneficial to economic development, often ignore the protection of ecosystems [18]. The impact of LUCC on soil degradation may seriously threaten environmental sustainability. The substantial benefits of LUCC on soil erosion control need to be further evaluated.

Various methods have been used to assess soil erosion by wind and water. With the development and comprehensive application of geographic information systems (GISs) and other technologies, statistical and empirical soil erosion modeling has been developed. For soil erosion by wind, the wind erosion equation (WEQ) [19], Texas erosion analysis models (TEAMs) [20], the Bocharov model [21], the revised wind erosion equation (RWEQ) [22], and the wind erosion prediction system (WEPS) [23] have been put forward successively. For soil erosion by water, the current models include the universal soil loss equation (USLE) [24], the revised universal soil loss equation (RUSLE) [25], Chinese soil loss equation (CSLE) [26], water erosion prediction project (WEPP) [27], LImburg soil erosion model (LISEM) [28], European soil erosion model (EUROSEM) [29], and so on. Among others, the revised wind erosion equation (RWEQ) and the revised universal soil loss equation (RUSLE) are the most popular models used to estimate the soil erosion modulus by wind and water, respectively [30,31]. However, long-term and large-scale monitoring of soil erosion processes remain a challenge.

The Qaidam Basin belongs to the sandy area of northern China in the soil erosion zone [32]. It is among the areas most sensitive to climate change in the entire Qinghai–Tibet Plateau [33]. The erosion environment in the area is complex and multiple erosion types coexist [34]. In addition, the climate of the Qaidam Basin tends to be warm and humid, with significant increases in temperature and rainfall brought on by global climate change [35]. Since the implementation of the Western development strategy, the population density of the Qaidam Basin has been rising, the economy and society of the entire region have been developing rapidly, and the intensity of regional land use has been increasing [36]. The higher intensity of land use has increased the risk of soil erosion by wind and water. Meanwhile, various ecological restoration and sand control projects, for example, the "Three North" Shelterbelt Development, Grain for Green, Partnership to Combat Land Degradation, and Grassland Ecological Protection Program, have been initiated [37]. However, how to determine the impact of these projects on soil erosion by wind and water in the Qaidam Basin remains unclear.

To better understand the impacts of LUCC on soil erosion by wind and water, we analyzed erosion dynamics in the Qaidam Basin from 2000 to 2018 and the contribution of LUCC to these changes using RWEQ and RUSLE. The ultimate goal of this study is to propose effective control measures to prevent soil erosion and strengthen local ecological protection and high-quality development.

## 2. Materials and Methods

### 2.1. Study Area

The Qaidam Basin is located on the northeastern edge of the Qinghai–Tibet Plateau, between 34°41′–39°20′ N and 87°48′–99°18′ E. It is the only large plateau inland basin in

the world, with elevations ranging from 2653 to 6748 m. As shown in Figure 1, the entire basin is generally triangular in shape and extends northwest–southeast, surrounded by the Altun, Qilian, and Kunlun Mountains, respectively.

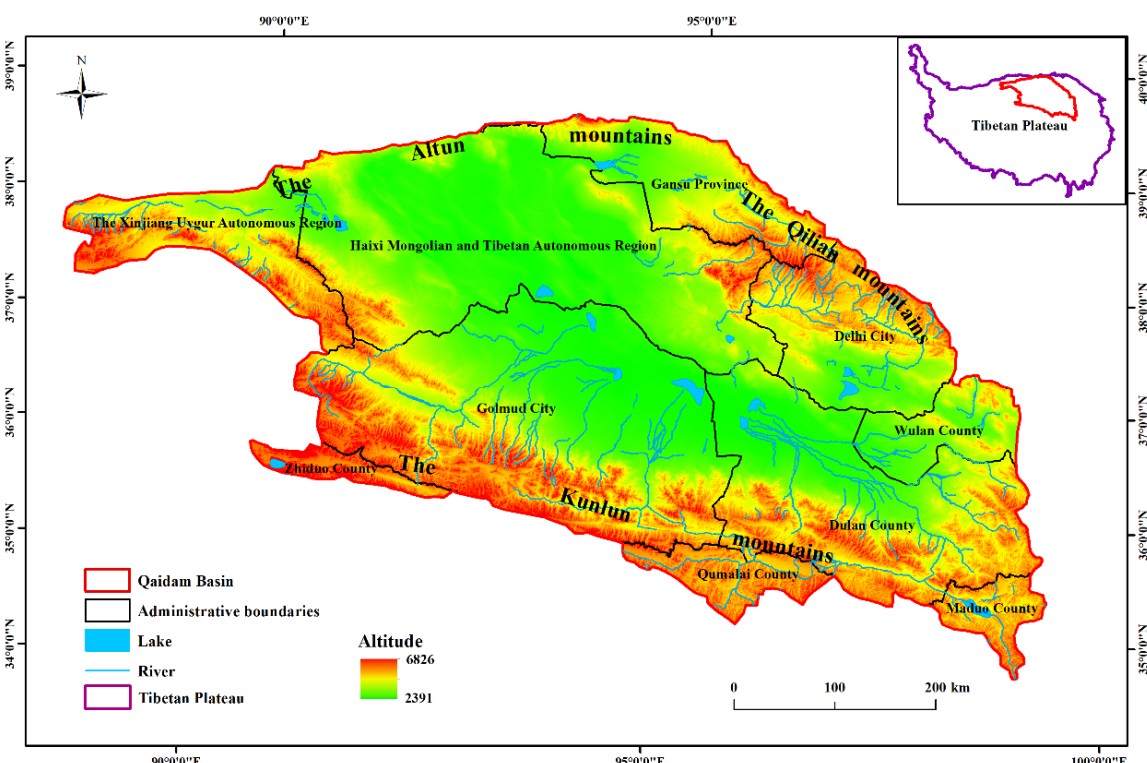

**Figure 1.** Geographical location of the Qaidam Basin.

The Qaidam Basin has a typical alpine, dry continental climate. The total area of the basin is approximately 276,500 km², mainly located in Qinghai, Xinjiang, and Gansu Provinces. Of these, 240,100 km² are in Qinghai Province, whereas 17,800 km² are in Xinjiang Province and 18,600 km² are in Gansu Province [38]. From 1961 to 2010, the mean annual rainfall was 82 mm, which decreased from east to west. The mean annual rainfall in the east is approximately 300 mm, whereas that in the west is approximately 20 mm. Affected by terrain and latitude, the mean annual temperature in the basin is approximately 3.5 °C, but its distribution is uneven, being high in the middle and low in the surrounding areas, high in the south, and low in the north. The mean annual sunshine time is more than 3000 h [33].

### 2.2. Modeling Soil Erosion Caused by Wind and Water

2.2.1. Modulus of Soil Erosion by Wind

The RWEQ model has been used to assess soil erosion by wind [30]. In this study, the modulus of the annual soil erosion by wind was obtained by first calculating monthly moduli and then adding them for 12 months. To eliminate the impacts of extreme weather, we used multi-year averaged monthly values during 1998–2002, 2008–2012, and 2014–2018 as inputs for 2000, 2010, and 2018. The RWEQ model was quantified using the formulas proposed by Fryrear et al. [22]:

$$SWEM_i = \frac{2000 \times Z}{S^2} \times Q_{max} \times e^{-\left(\frac{Z}{S}\right)^2} \tag{1}$$

$$S = 150.71 \times \left(WF \times EF \times SCF \times K' \times C_{RWEQ}\right)^{-0.3711} \tag{2}$$

$$Q_{max} = 109.8 \times \left( WF \times EF \times SCF \times K' \times C_{RWEQ} \right) \tag{3}$$

where $SWEM_i$ is the mean modulus of soil erosion by wind in month i [t/(km$^2$·a)]; S is the key block length (m); $Q_{max}$ is the maximum amount of soil transport (kg/m); Z is the maximum wind erosion distance in the downwind direction (m), set to 50 m [39]; WF is the weather factor (kg/m) calculated using a function of the wind factor (Wf), soil wetness factor (SW), and snow depth factor (SD); EF and SCF represent the soil erodible and soil crust factors, respectively; K' is the soil roughness factor extracted by the digital elevation model (DEM); and $C_{RWEQ}$ is the vegetation factor. The EF, SCF, K', and $C_{RWEQ}$ values were dimensionless.

The WF is calculated as follows [40]:

$$WF = Wf \times SW \times SD \times \frac{\rho}{g} \tag{4}$$

$$Wf = u_2 \times (u_2 - u_1)^2 \times N_d \tag{5}$$

$$SW = \frac{ET_P - (R + I) \times \frac{R_d}{N_d}}{ET_P} \tag{6}$$

$$ET_P = 0.0162 \times \left( \frac{SR}{58.5} \right) \times (DT + 17.8) \tag{7}$$

$$SD = 1 - P \tag{8}$$

where Wf is the wind factor (m$^3$/s$^3$); SW is the soil wetness factor (dimensionless); SD is the snow cover factor (dimensionless); g is the acceleration due to gravity (m/s$^2$); $\rho$ is the air density (kg/m$^3$); $u_2$ is the wind speed at a height of 2 m (m/s); $u_1$ is the threshold wind velocity at a height of 2 m, which is 5 m/s in this study; $N_d$ is the number of days when the wind speed is greater than 5 m/s in each month; R is the rainfall in each month (mm); I is the amount of irrigation in each month (mm); $R_d$ is the number of days of rainfall or irrigation in each month; $ET_p$ is the potential evapotranspiration in each month (mm); DT is the mean temperature in each month (°C); SR is the total solar radiation in each month (cal/cm$^2$); and P is the probability that the depth of snow cover is greater than 25.4 mm in each month.

The EF and SCF are expressed as follows [41]:

$$EF = \frac{29.09 + 0.31 \times sa + 0.17 \times si + 0.33 \times \frac{sa}{cl} - 2.59 \times OM - 0.95 \times CaCO_3}{100} \tag{9}$$

$$SCF = \frac{1}{1 + 0.0066 \times cl^2 + 0.021 \times OM^2} \tag{10}$$

where $CaCO_3$, OM, sa, si, and cl represent the calcium carbonate, soil organic matter, sand, silt, and clay content (%), respectively.

K' is expressed as follows [42]:

$$K' = \cos \alpha \tag{11}$$

where $\alpha$ represents the slope gradient.

The vegetation factor ($C_{RWEQ}$) is expressed as follows [39]:

$$C_{RWEQ} = e^{-0.0483 \times SC} \tag{12}$$

where SC is the vegetation coverage in each month (%).

2.2.2. Modulus of Soil Erosion by Water

RUSLE is used for large-scale soil erosion by water assessments worldwide [31]. To eliminate the impacts of extreme weather, we obtained the modulus of the annual soil water erosion using a variable control method with multi-year averaged monthly values during 1998–2002, 2008–2012, and 2014–2018 as inputs for 2000, 2010, and 2018. The RUSLE model was calculated using the formula proposed by Renard et al. [25]:

$$A = 100 \times R \times K \times LS \times C_{RUSLE} \times P \tag{13}$$

where A is the mean modulus of the annual soil erosion by water (t/(km$^2$·a)); R is the rainfall erosivity factor (MJ·mm/(hm$^2$·h·a)); K is the soil erodible factor (t·h/(hm$^2$·MJ·mm)); LS is the slope length and steepness factor; C$_{RUSLE}$ is the coverage and management factor; and P is the soil conservation measures factor. The LS, C$_{RUSLE}$, and P values were dimensionless. The unit conversion factor is 100.

R could be calculated as suggested by Wischmeier [43]:

$$R = \sum_{i=1}^{12} 1.735 \times 10^{\left(1.5 \times \lg \frac{Pi^2}{P} - 0.8188\right)} \tag{14}$$

where p is annual average precipitation (mm); and pi is monthly precipitation (mm).

K could be calculated according to Sharply and Willians [44]:

$$K = \left\{ 0.2 + 0.3 \times \exp\left[ -0.0256 \times sa\left(1 - \frac{si}{100}\right) \right] \right\} \times \left[ \frac{si}{cl+si} \right]^{0.3} \times \left\{ 1 - 0.25 \times \frac{C}{C+\exp(3.72-2.95\times C)} \right\}$$
$$\times \left\{ 1 - 0.7 \times \frac{Sn}{Sn+\exp(22.9\times Sn-5.51)} \right\} \tag{15}$$

$$Sn = 1 - \frac{sa}{100} \tag{16}$$

where C represents the soil organic carbon content; and sa, si, and cl represent the sand, silt, and clay content (%), respectively, having the same meaning as in Equation (10).

LS is expressed as follows [45,46]:

$$L = \left( \frac{\lambda}{22.1} \right)^m \tag{17}$$

$$m = \begin{cases} 0.2 & \theta < 1° \\ 0.3 & 1° \leq \theta < 3° \\ 0.4 & 3° \leq \theta < 5° \\ 0.5 & \theta \geq 5° \end{cases} \tag{18}$$

$$S = \begin{cases} 10.8 \times \sin\theta + 0.036 & \theta < 5° \\ 16.8 \times \sin\theta - 0.05 & 5° \leq \theta < 10° \\ 21.9 \times \sin\theta - 0.96 & \theta \geq 10° \end{cases} \tag{19}$$

where L is the slope length factor; S is the slope steepness factor; λ is the slope length (m); 22.1 is the standard plot length; m is the slope length index (dimensionless); and θ is the slope (°).

C$_{USLE}$ could be calculated following Cai et al. [47]:

$$C_{USLE} = \begin{cases} 1 & c = 0 \\ 0.6508 - 0.3436 \times \lg_c & 0 < c \leq 78.3\% \\ 0 & c > 78.3\% \end{cases} \tag{20}$$

where c is the monthly average vegetation coverage (%).

In this study, the values of the soil conservation measures factor (P) were assigned according to Table 1 [48]:

**Table 1.** Soil conservation measures factor (P) values under different land uses.

| Land Use | Cropland | Woodland | Grassland | Waters | Construction Land | Unused Land |
|----------|----------|----------|-----------|--------|-------------------|-------------|
| P | 0.35 | 1 | 1 | 0 | 0 | 1 |

### 2.2.3. Data Sources

We obtained information regarding the five factors of the RWEQ model (WF, EF, SCF, K′, and $C_{RWEQ}$) and five factors of the RUSLE model (R, K, LS, $C_{RUSLE}$, and P) that included data about geographic background, meteorological and hydrological observations, vegetation coverage, and field research studies (Table 2). These data were of various types and formats; therefore, the uniform projection method (Albers ellipsoid) and data accuracy (100 m) using ArcGIS 10.7 (https://www.esri.com, accessed on 1 March 2021) were required to build the database of erosion by wind and water in the Qaidam Basin.

**Table 2.** Data collected in this study.

| Data Types | Temporal Resolution | Spatial Resolution | Format | Source | Application |
|------------|---------------------|---------------------|--------|--------|-------------|
| Land use | 2000, 2010, 2018 | 100 m | Grid | http://www.resdc.cn (accessed on 1 March 2021) | LUCC analysis |
| DEM | 2009 | 30 m | Grid | http://www.gscloud.cn (accessed on 1 March 2021) | LS and K′ |
| Wind speed | 3 h (1 January 1998–31 December 2018) | 0.1° | Grid | http://data.tpdc.ac.cn (accessed on 1 March 2021) | WF |
| Temperature | Day (1 January 1998–31 December 2018) | Weather station | Text | Haixi Meteorological Bureau (accessed on 1 March 2021) | WF |
| Rainfall | Day (1 January 1998–31 December 2018) | Weather station | Text | Haixi Meteorological Bureau (accessed on 1 March 2021) | WF and R |
| Solar radiation | Day (1 January 1998–31 December 2018) | 0.1° | Grid | http://data.tpdc.ac.cn (accessed on 1 March 2021) | WF |
| Snow depth | Day (1 January 1998–31 December 2018) | 25 km | Grid | http://data.tpdc.ac.cn(accessed on 1 March 2021) | WF |
| SPOT-NDVI | Month (2000, 2010, 2018) | 1000 m | Grid | http://www.resdc.cn (accessed on 1 March 2021) | $C_{RWEQ}$ and $C_{RUSLE}$ |
| Soil data (texture, organic matter, calcium carbonate) | 2009 | 1:1 million | Grid | http://data.tpdc.ac.cn (accessed on 1 March 2021) | EF, SCF, and K |

### 2.3. Classification Standards of Soil Erosion by Wind and Water

The grades of soil erosion by wind and water were expressed according to the soil erosion grade standard (SL 190-2008), as shown in Table 3.

**Table 3.** Classification of soil wind erosion and water erosion moduli.

| Class | Wind Erosion (t/(km$^2$·a)) | Water Erosion (t/(km$^2$·a)) |
|-------|------------------------------|-------------------------------|
| Tolerable erosion | <200 | <500 |
| Slight erosion | 200–2500 | 500–2500 |
| Moderate erosion | 2500–5000 | 2500–5000 |
| Severe erosion | 5000–8000 | 5000–8000 |
| Very severe erosion | 8000–15,000 | 8000–15,000 |
| Destructive erosion | >15,000 | >15,000 |

### 2.4. Analysis of the LUCC Impact on Soil Erosion by Wind and Water

According to the secondary classification system of the China Multi-period Land Use/Land Cover Change Remote Sensing Monitoring Database, the land considered in this study included cropland, woodland (dense woodland, sparse woodland, and other

woodland), shrubbery, grassland, water, construction land, desert, and other unused land (Gobi, saline-alkali land, marshland, bare land, bare rock texture, etc.) (http://www.resdc.cn, accessed on 1 March 2021). We compared the moduli of average soil erosion by wind and water for cropland, grassland, woodland, shrubbery, desert, and other unused lands in 2000, 2010, and 2018 to examine the effects of land use type. ArcGIS 10.7 was used to obtain land use transfer matrices for 2000–2010, 2010–2018, and 2000–2018. Land use transfer maps were created using Origin Pro 2021b (https://www.originlab.com, accessed on 1 March 2021).

To further highlight the main effects of LUCC on soil erosion, a variable control method was used to eliminate climatic effects by taking multi-year averaged monthly values from 2000 to 2018 as inputs [12]. The net changes in the moduli of soil erosion by wind (Equation (21)) and water (Equation (22)) due to LUCC were analyzed by superimposing the annual moduli of average soil erosion by wind and water and LUCC maps as follows:

$$\mathrm{SWEC} = \sum_{m=1}^{n} \mathrm{area}_m \left( \mathrm{swec}_j - \mathrm{swec}_k \right) \qquad (21)$$

$$\mathrm{AC} = \sum_{m=1}^{n} \mathrm{area}_m \left( \mathrm{ac}_j - \mathrm{ac}_k \right) \qquad (22)$$

where SWEC is the net change in the mean modulus of soil erosion by wind (t/(km$^2$ × a)); AC is the net change in the mean modulus of soil erosion by water (t/(km$^2$ × a)); n is the total number of LUCC types; m is the total number of mth LUCC types; area$_m$ is the area of the mth LUCC type; and j and k are the end and start years of the study period, respectively.

## 3. Results

### 3.1. Temporal and Spatial Changes in LUCC

In 2000, 2010, and 2018, the land use structure of the Qaidam Basin was dominated by other unused land, grassland, and desert, which together accounted for 96.61%, 96.17%, and 96.07% of the total area, respectively, with grassland accounting for 29.92%, 30.45%, and 31.39% and desert accounting for 12.30%, 12.32%, and 12.24%, respectively. There was great consistency in the spatial distribution of cropland, woodland, and construction land concentrated in the eastern region (Delingha, Dulan, Golmud, and Ulan). The deserts are concentrated in the western region and Dulan County. Grasslands and other unused land are evenly distributed in the Qaidam Basin (Figure 2).

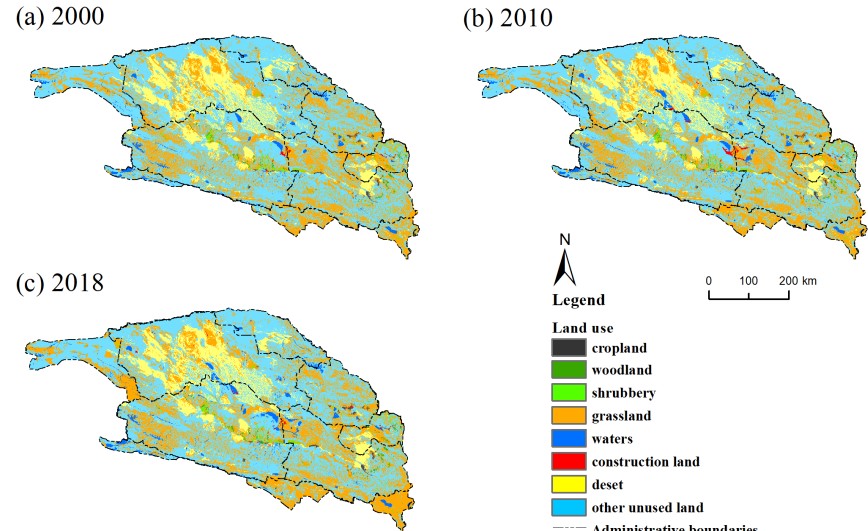

**Figure 2.** Distribution of different land use types from 2000 to 2018 in the Qaidam Basin ((**a**): in 2000. (**b**): in 2010. (**c**): in 2018).

From 2000 to 2018, the woodland, cropland, watershed, and grassland areas increased by 30.47 km², 53.84 km², 1317.34 km², and 4083.26 km², respectively. The area of other unused lands continued to decrease by 106.36 km². The area of shrubbery decreased by 39.76 km² from 2000 to 2010 and then increased by 33.79 km² from 2010 to 2018. From 2000 to 2010, the areas of desert and construction land increased by 50.41 km² and 444.38 km², respectively, and then decreased by 222.02 km² and 338.02 km² from 2010 to 2018. From 2000 to 2018, the area with land use conversion comprised 35,993.41 km², i.e., 12.96% of the total area. The cropland, woodland, shrubbery, and other unused land areas that were turned to grassland comprised 47.38%, 46.40%, 55.45%, and 79.84% of their total areas that underwent conversion, respectively. In addition, 206.37 km² of grassland and other unused land were converted into woodland, which became the main source of woodland growth, mainly in Delingha, Ulan, Dulan, and Golmud. However, there was an increase in other unused land of 13,201.72 km², with the largest area of grassland having been turned into other unused land. Furthermore, 2771.40 km² of land was transformed into desert, with the largest contribution of other unused land being mainly in the western region and in parts of Golmud, Dulan, Ula, and Delingha (Figures 3 and 4).

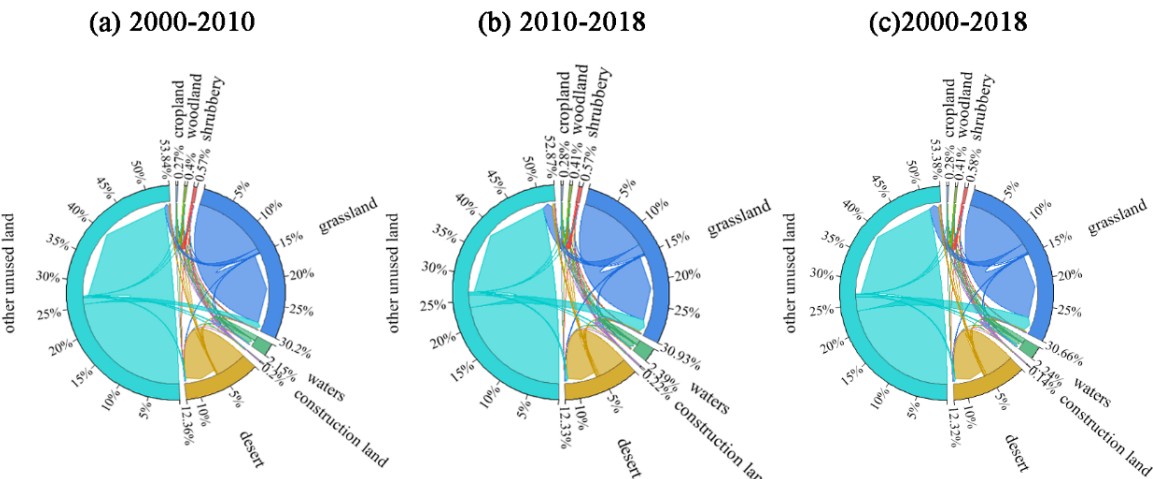

**Figure 3.** The land use shifts in the Qaidam Basin ((**a**): 2000–2010. (**b**): 2010–2018. (**c**): in 2000–2018).

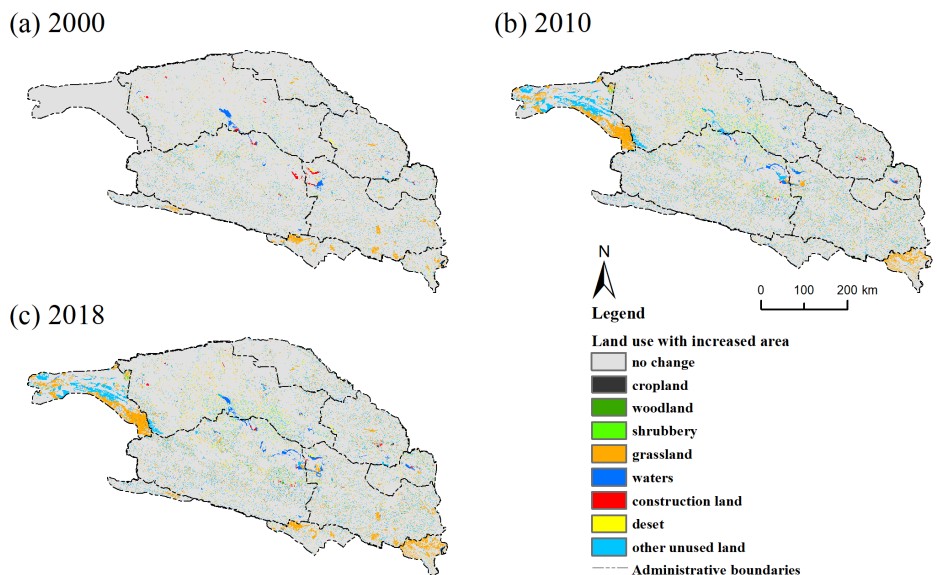

**Figure 4.** Regional distribution map of land use with increased area in the Qaidam Basin ((**a**): in 2000. (**b**): in 2010. (**c**): in 2018).

*3.2. Temporal and Spatial Changes of Soil Erosion by Wind and Water*

3.2.1. Erosion by Wind

From 2000 to 2018, the area affected at least to some extent by wind erosion in the Qaidam Basin was 249,437.37 km², occupying 89.79% of the total area. Slight and destructive erosion were the main types of soil erosion by wind, which covered 84,214.48 km² and 83,136.48 km², i.e., 33.76% and 33.33%, respectively, of the total area affected by wind erosion. Spatially, the modulus of soil erosion by wind was low in the east and high in the west. Areas with moderate erosion and above were mainly concentrated in the Haixi Mongolian and Tibetan Autonomous Prefectures in the Qaidam Basin (Figure 5a), where wind erosion hazards are relatively severe.

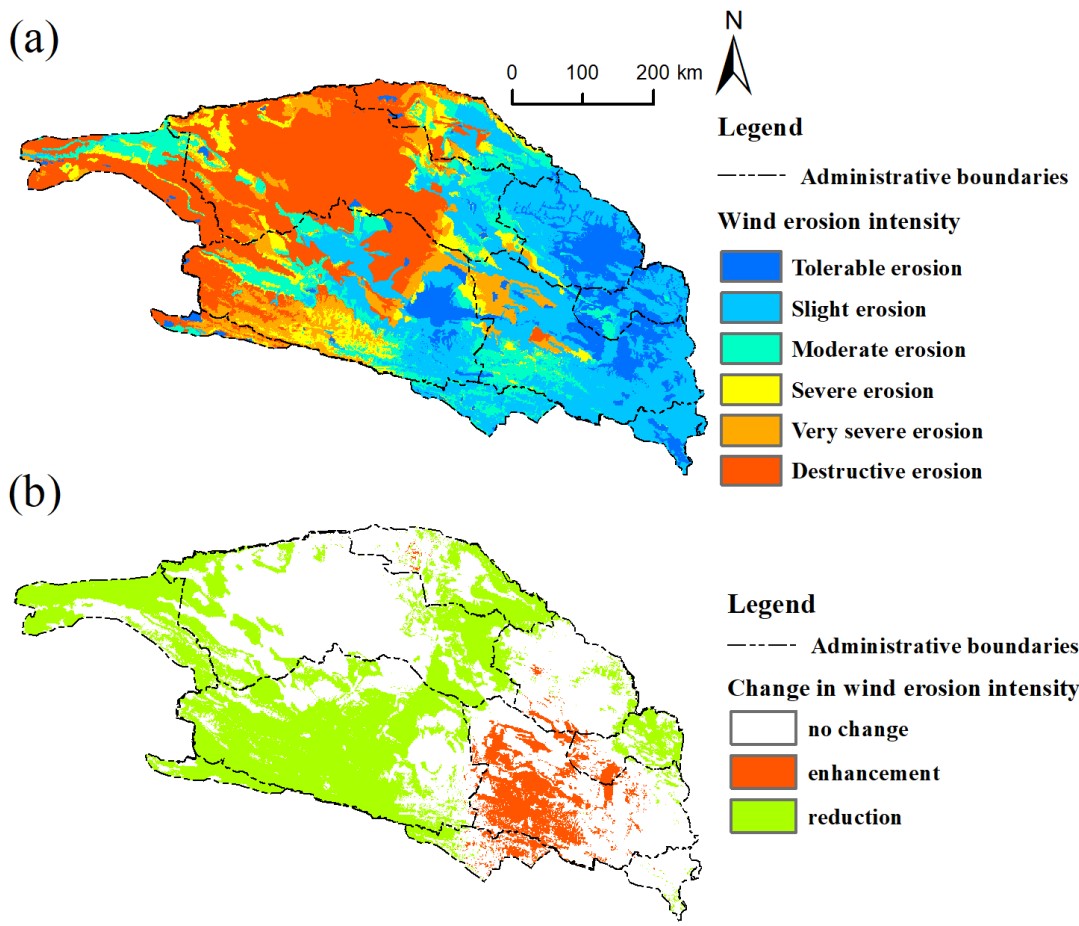

**Figure 5.** Spatial distribution (**a**) and spatial variation (**b**) of the soil wind erosion intensity from 2000 to 2018 in Qaidam Basin.

From 2000 to 2018, the extent of total erosion by wind in the Qaidam Basin decreased by $2.12 \times 10^8$ tons. The area affected by slight erosion and above decreased by 6884.62 km²; it mainly encompassed Golmud City, Haixi Mongolian and Tibetan Autonomous Prefecture in Qinghai Province, and Ruoqiang County in Xinjiang Province (Figure 5b). The intensity of erosion by wind also decreased, and 60.36%, 80.14%, 90.66, and 90.07% of areas affected by slight, moderate, severe, and very severe wind erosion changed into areas with correspondingly lower erosion degrees. However, the intensity of erosion tended to strengthen locally, mainly in the Dulan area (Figure 5b). The area with enhanced erosion intensity was 18,682.89 km², of which the areas with slight and moderate erosion were mostly transformed into those with higher grades of erosion, accounting for 59.34% of the area with enhanced erosion intensity (Table 4).

**Table 4.** The transition matrix of soil wind erosion intensity classification in Qaidam Basin from 2000 to 2018 (area: km$^2$).

| 2000 | 2018 | | | | | | |
|---|---|---|---|---|---|---|---|
| | **Tolerable** | **Slight** | **Moderate** | **Severe** | **Very Severe** | **Destructive** | **Total** |
| Tolerable | 22,753.06 | 3235.27 | 0 | 0 | 0 | 0 | 25,988.33 |
| Slight | 10,112.57 | 55,290.66 | 6124.39 | 478.70 | 37.38 | 0 | 72,043.70 |
| Moderate | 7.32 | 17,935.71 | 8104.00 | 3004.78 | 1423.95 | 17.33 | 30,493.09 |
| Severe | 0 | 11,829.45 | 5227.82 | 2094.52 | 1018.92 | 739.25 | 20,909.96 |
| Very severe | 0 | 6925.69 | 11,445.09 | 5227.94 | 4796.87 | 2602.92 | 30,998.51 |
| Destructive | 0 | 194.98 | 6096.04 | 9642.78 | 20,641.59 | 60,806.51 | 97,381.90 |
| total | 32,872.95 | 95,411.76 | 36,997.34 | 20,448.72 | 27,918.71 | 64,166.01 | 277,815.49 |

### 3.2.2. Erosion by Water

From 2000 to 2018, the area influenced by water erosion to any degree in the Qaidam Basin was 109,630.20 km$^2$, occupying 39.63% of the total area. Slight erosion was mainly observed, covering an area of 69,454.11 km$^2$, occupying 63.35% of the total area affected by water erosion. The area where erosion was moderate and above covered 40,176.09 km$^2$, of which the area of moderate erosion was the largest, accounting for 23.60% of the total area with any erosion by water. As the intensity of erosion increased, the erosion area became smaller and smaller. The spatial distribution of soil erosion by water showed a trend opposite to that of erosion by wind, with a high intensity of erosion by water in the east and a gradual decrease in the west. Areas affected by moderate and above levels of erosion were concentrated in the Qilian and Kunlun Mountains, where water erosion hazards were relatively severe (Figure 6a).

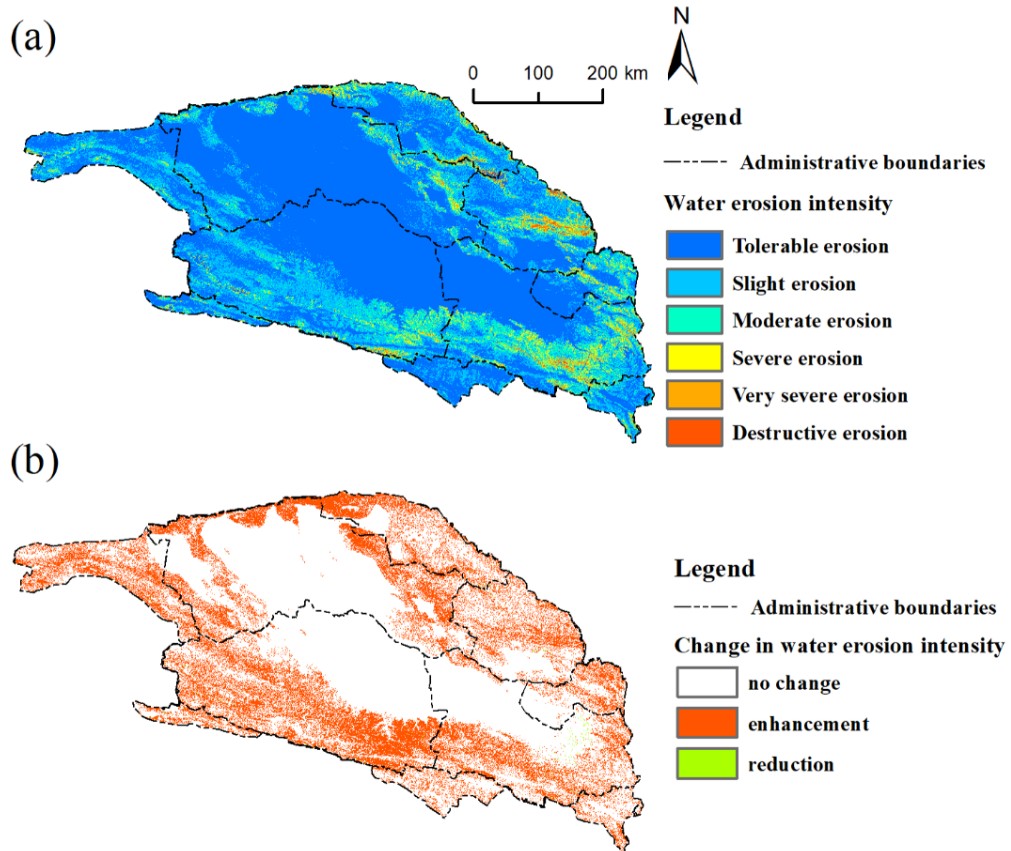

**Figure 6.** Spatial distribution (**a**) and spatial variation (**b**) of the soil water erosion intensity from 2000 to 2018 in Qaidam Basin.

From 2000 to 2018, erosion by water in the Qaidam Basin gradually increased by $2.07 \times 10^8$ tons. The area affected by water erosion increased by 32,486.46 km$^2$, mainly in the Altun, Qilian, and Kunlun Mountains (Figure 6b). The intensity of erosion also increased, and 99.31%, 98.72%, 97.49%, and 94.54% of areas that had light, moderate, heavy, and very heavy extents of erosion by water, respectively, became areas with higher erosion grades. However, there remained small areas where the intensity of erosion by water showed a weakening trend, but those were small and scattered. The total area in which the intensity of erosion by water became lower was 472.63 km$^2$, and 75.36% of it comprised regions with light and moderate erosion (Table 5).

**Table 5.** The area transition matrix of soil water erosion intensity classification in Qaidam Basin from 2000 to 2018 (km$^2$).

| 2020 | 2018 | | | | | | |
|---|---|---|---|---|---|---|---|
| | **Tolerable** | **Slight** | **Moderate** | **Severe** | **Very Severe** | **Destructive** | **Total** |
| Tolerable | 15,5766.35 | 32,363.18 | 194.12 | 84.82 | 124.52 | 68.66 | 18,8601.65 |
| Slight | 207.74 | 36,227.60 | 23,849.32 | 4773.38 | 1015.54 | 54.19 | 66,127.77 |
| Moderate | 75.90 | 72.54 | 5306.82 | 7891.15 | 3161.67 | 374.91 | 16,882.99 |
| Severe | 37.40 | 0.12 | 34.32 | 980.58 | 2377.00 | 408.17 | 3837.59 |
| Very severe | 24.14 | 0 | 0 | 13.90 | 393.20 | 658.71 | 1089.95 |
| Destructive | 3.66 | 0 | 0 | 0 | 2.91 | 100.66 | 107.23 |
| total | 156,115.19 | 68,663.44 | 29,384.58 | 13,743.83 | 7074.84 | 1665.30 | 27,6647.18 |

*3.3. Contribution of LUCC to Soil Erosion by Wind and Water*

Water and construction lands were assumed to be free of any form of erosion by wind or water in the calculation, so no comparisons of the moduli of soil erosion by wind and water were performed for these two land types.

The modulus of soil erosion by wind depended on the land use as follows: desert (80,404.6 t/(km$^2$·a)) > grassland (20,893.45 t/(km$^2$·a)) > other unused land (16,428.5 t/(km$^2$·a)) > woodland (2232.52 t/(km$^2$·a)) > shrubbery (1559.41 t/(km$^2$·a)) > cropland (347.82 t/(km$^2$·a)). The change in the type of land use resulted in an increase of $2.61 \times 10^7$ tons, which is manifested as a decrease of $1.50 \times 10^6$ tons and an increase of $2.76 \times 10^7$ tons in soil erosion by wind. (Figure 7). The conversion of other unused land to grassland accounted for 96.37% of the reduction, mainly in the Qilian and Kunlun Mountains in Qinghai and Ruoqiang County in Xinjiang (Figure 8). The transformation of other unused land and grassland to desert, grassland to other unused land, and desert to grassland and other unused land contributed 99.95% of the increase, mainly in the western sandy area (Figure 8). It is important to note that between 2010 and 2018, the land use types remained unchanged, but the vegetation coverage changes resulted in an increase of $3.29 \times 10^8$ tons, which is manifested as a decrease of $2.66 \times 10^4$ tons and an increase of $3.295 \times 10^8$ tons in soil erosion by wind. Cropland contributed to the reduction in wind erosion, whereas desert, other unused land, and grassland contributed 46.10%, 27.75%, and 26.12% of the increase (Figure 7), respectively. In summary, land use/cover changes resulted in an increase of $3.55 \times 10^8$ tons and were negative to the soil erosion of wind.

The modulus of soil erosion by water was also dependent on the type of land use: other unused land (1515.42 t/(km$^2$·a)) > woodland (1313.75 t/(km$^2$·a)) > shrubbery (874.55 t/(km$^2$·a)) > grassland (18.47 t/(km$^2$·a)) > desert (102.16 t/(km$^2$·a)) > cropland (88.60 t/(km$^2$·a)). The change in the type of land use resulted in a decrease of $2.47 \times 10^6$ tons, which is manifested as a decrease in wind erosion of $2.50 \times 10^6$ tons and an increase of $2.54 \times 10^4$ tons in soil erosion by water (Figure 9). The conversion of other unused land to grassland, woodland to grassland, grassland to woodland, and grassland and shrubbery to other unused land contributed 92.20% of the reduction, mainly in the Qilian and Kunlun Mountains, especially in the former (Figure 10). The conversion of cropland to other unused land and grassland contributed 98.41% of the increase, which was consistent with the concentrated distribution

of cropland in the eastern region of the Qaidam Basin (Delingha, Dulan, Golmud, and Ulan) (Figure 10). Importantly, the land use types remained unchanged but the changes in vegetation coverage reduced soil water loss by $1.60 \times 10^7$ tons, mainly owing to other unused land and grassland, which contributed 67.27% and 28.90% of the decrease (Figure 9), respectively. In summary, land use/cover changes resulted in a decrease of $1.85 \times 10^8$ tons and were positive to the soil erosion of water.

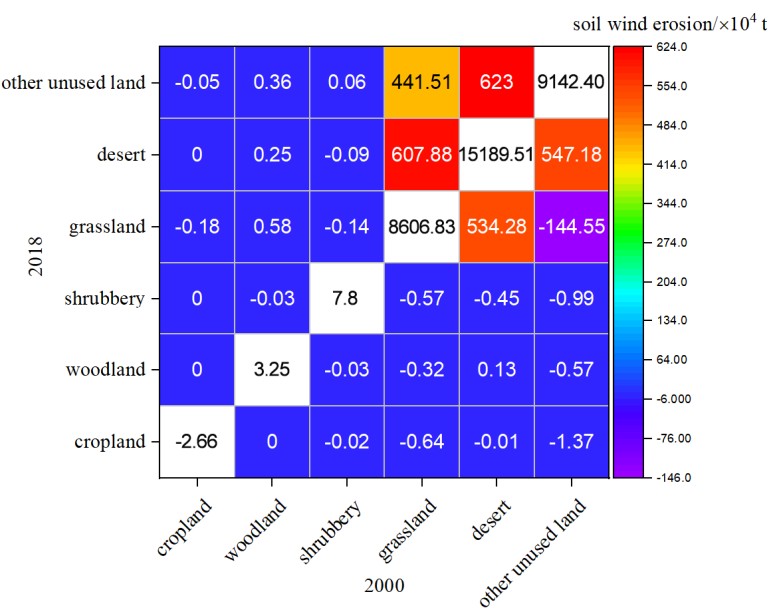

**Figure 7.** Changes in soil wind erosion caused by the conversion of land use types from 2000 to 2018 in Qaidam Basin.

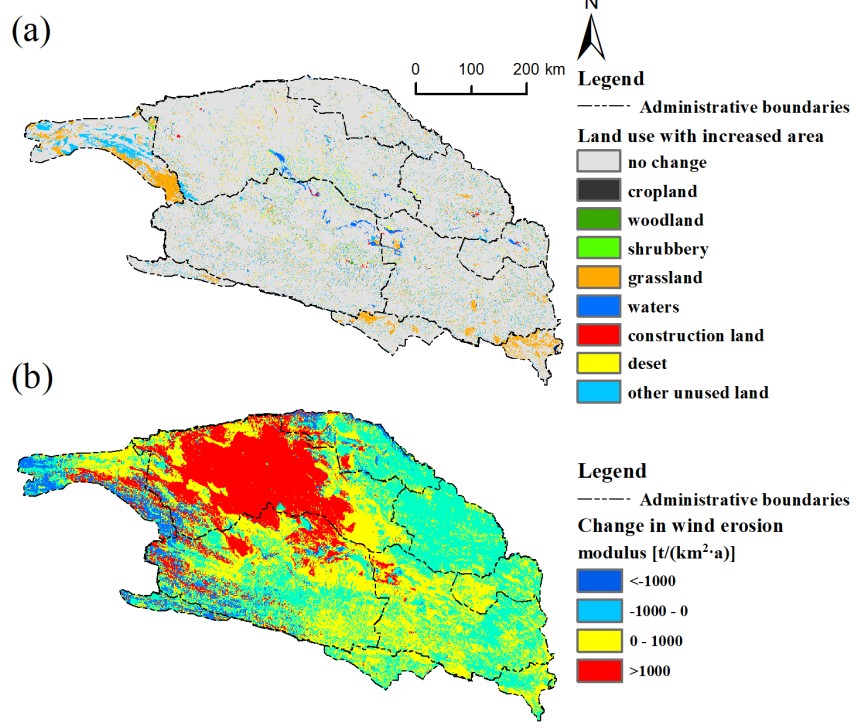

**Figure 8.** Spatial distribution of land use types with increased area (**a**) and corresponding soil wind erosion changes (**b**) from 2000 to 2018 in Qaidam Basin.

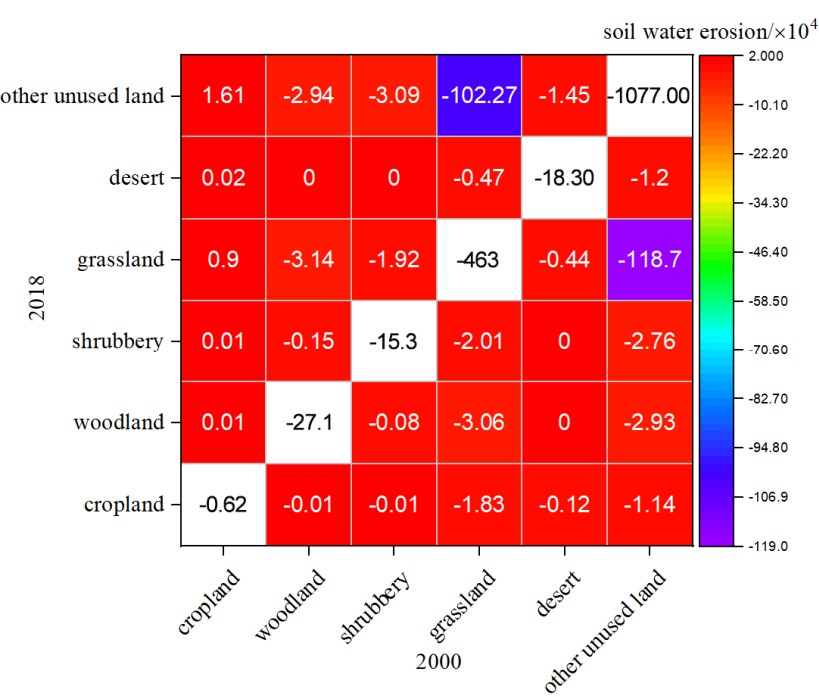

**Figure 9.** Changes in soil water erosion caused by the conversion of land use types from 2000 to 2018 in Qaidam Basin.

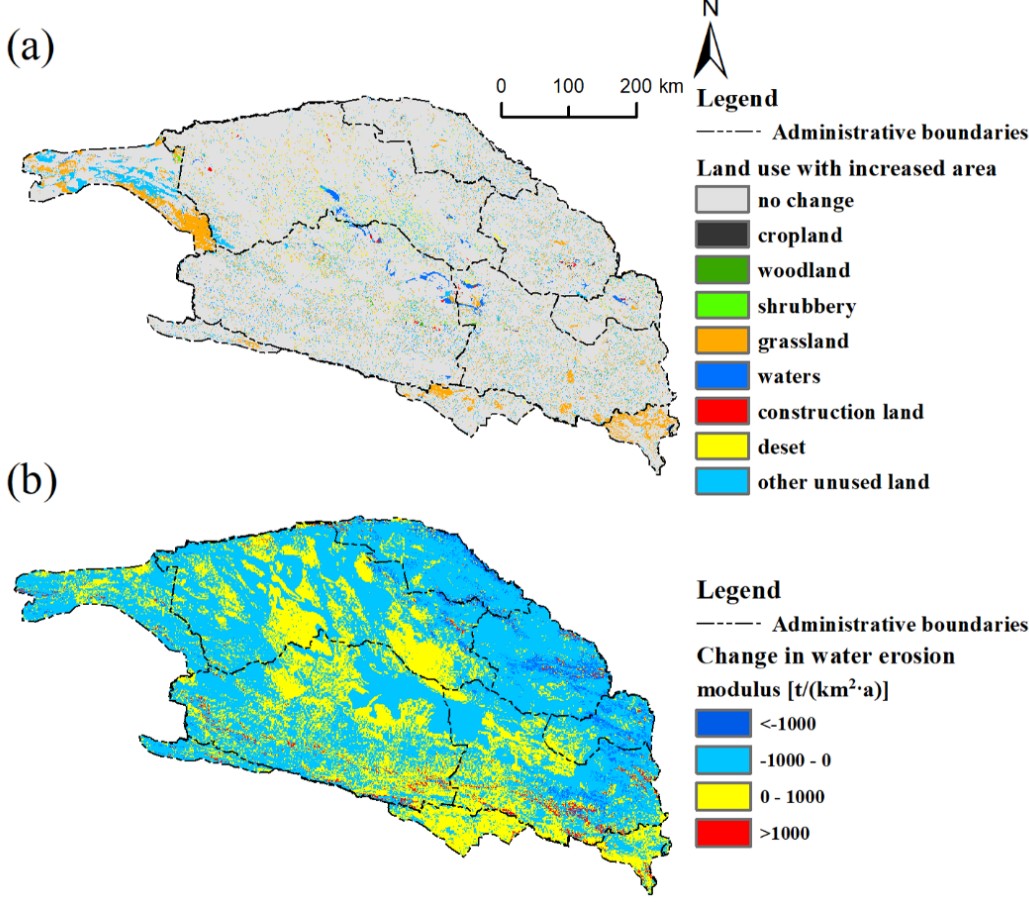

**Figure 10.** Spatial distribution of land use types with increased area (**a**) and corresponding soil water erosion changes (**b**) from 2000 to 2018 in Qaidam Basin.

## 4. Discussion

### 4.1. Current Status and Changing Characteristics of Soil Erosion by Wind and Water in Qaidam Basin

The Qaidam Basin is located in the sandy area of northern China. The main type of erosion there is erosion by wind, with some erosion by water occurring after heavy rains [32]. In this study, the modulus of the average erosion by wind calculated for the Qaidam Basin in 2000, 2010, and 2018 was 31,183.70, 24,705.20 and 23,547.60 t/(km$^2$·a), with an average of 26,478.83 t/(km$^2$·a). It is close to the mean value of 57,990 t/(km$^2$·a) calculated by Wang et al. [49] based on the RWEQ model for the actual wind erosion in Mangya City in the Qaidam Basin, and Mangya City is located in the Haixi Mongolian and Tibetan Autonomous Prefecture, which is a region that suffers severe wind erosion. The area of moderate erosion (2500–5000 t/(km$^2$·a)) and above in the Qaidam Basin was 165,222.89 km$^2$, i.e., 59.47% of the total area. Jiang et al.'s study also confirmed that wind erosion zones above the moderate level in Qinghai Province are mainly located in areas such as the periphery of the Qaidam Basin [39]. The average erosion by water modulus calculated for the Qaidam Basin in 2000, 2010, and 2018 was 692.97, 1203.38, and 1442.47 t/(km$^2$·a), with an average of 1112.94 t/(km$^2$·a), which was considered as slight erosion. The area of slight erosion (500–2500 t/(km$^2$·a)) in the Qinghai part of the Qaidam Basin in 2019 accounted for 86.5% of the total area affected by water-caused erosion [50]. These results indicate that the calculated moduli of soil erosion by wind and water were reliable. However, the limitation of our study was that the RWEQ model did not consider the impact of engineering measures on the modulus of soil erosion by wind. Many sand control projects have been implemented in the Qaidam Basin, such as nylon mesh, stone, and woven bag sand barriers, and other engineering measures, which were all effective in intercepting wind and sand [51]. Therefore, the soil erosion by wind modulus in this study only reflects the erosion caused by vegetation changes.

From 2000 to 2018, the situation with the erosion caused by wind in the Qaidam Basin improved to some extent, owing to the reduction in the area affected by erosion and lower erosion intensity (Table 4). Data reported by Teng et al. [52], which showed that erosion by wind in the Qaidam Basin decreased considerably from 1980 to 2015, are in agreement with our results. Several locations in which erosion increased from tolerable, slight, or moderate to that with a higher intensity occupied 76.66% of the total area with increased intensity of erosion by wind, mainly in the Dulan area, which needs to be protected. At the same time, erosion by water in the Qaidam Basin intensified to some extent, and the erosion intensity level increased (Table 5). In particular, the areas in which tolerable or low erosion changed to that with a higher intensity occupied 80.79% of the total area with an increased degree of erosion by water, mainly in the Altun, Qilian, and Kunlun Mountains. This observation is mainly related to the increasing trend of annual rainfall in the Qaidam Basin from 2000 to 2018 [35]. In addition, the area with aggravated compound erosion by wind and water, mainly located in Dulan, comprised 4832.49 km$^2$ (Figure 11). Therefore, with climate change, not only should we strengthen the prevention and control of the types of erosion caused by wind and water separately, but we should also pay attention to the comprehensive management of the compound erosion by wind and water.

### 4.2. Characteristics of LUCC and Its Influencing Factors

In addition to natural factors, LUCC is the main driver of global environmental change [53]. Therefore, it is essential to explore the impacts of LUCC on the intensity of erosion by water and wind. The main types of LUCC in the Qaidam Basin are mainly caused by changes in the natural environment, social development, and implementation of ecological projects. In our study, we found an increasing trend in vegetation coverage in Qaidam from 2000 to 2018. In particular, areas occupied by grassland, woodland, and shrubs increased by 16.65%, 11.91%, and 3.71%, respectively. It has been shown that rising temperature, increased precipitation, and ecological construction have increased vegetation cover on the Qinghai–Tibet Plateau [54]. At the same time, with increasing population

density and rapid urban expansion in the Qaidam Basin, there has been a rapid change in land use types since 2000, with cropland, woodland, and grassland expansion and concomitant decreases in shrubbery, desert, and other unused lands. Although the total amount of cropland increased, the actual changes in the cropland were complex. The former cropland was converted into grassland, other unused land, and construction land, covering an area of 87.43 km$^2$, which may be related to the overall threat of desertification and the negative effect of salinization caused by the rising water table in Golmud and other places, resulting in the abandonment of cropland. At the same time, 133.05 km$^2$ of grassland and other unused land were replaced by new cropland, probably due to the reclamation of the desert in Golmud and other places for plantations of *Lycium chinense*. As of 2020, the area occupied by *Lycium chinense* plants in the Qaidam Basin exceeded $3.00 \times 10^6$ km$^2$ [55].

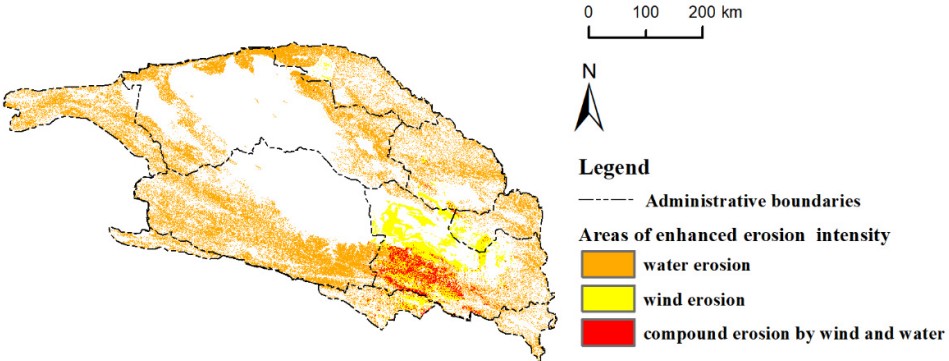

**Figure 11.** Spatial distribution of areas with enhanced wind and water erosion intensity.

Since 2000, China has implemented a large-scale project to transform degraded cropland and wasteland into woodland and grassland and increase vegetation cover [56,57], aiming to protect the local environment [58]. Under the guidance of the land use policy of ecological engineering, woodlands and grasslands have been restored and the situation with desertification has improved. In 2018, the areas occupied by woodlands, shrubbery, and grasslands in the Qaidam Basin have increased by 4113.96 km$^2$, and the conversion of other unused lands into woodlands, shrubbery, and grasslands was approximately 1.42-fold higher than the extent of the opposite change (Figure 3).

### 4.3. The Impact of LUCC on Erosion by Wind and Water

The increase in grassland in the Qaidam Basin was found to contribute to a reduction in soil erosion by wind and water. From 2000 to 2018, converting other unused land to grassland reduced wind-caused soil erosion by approximately $144.55 \times 10^4$ tons, occupying 96.37% of the total reduction in soil erosion by wind owing to LUCC (Figure 7). This may be related to the fact that the Chinese government has launched ecological projects to reduce wind erosion in northern China. Chi et al. [59] showed a decreasing trend of 0.71 t ha$^{-1}$ year$^{-1}$ in the wind erosion modulus because of ecological projects during 2000–2010. When desert–other unused land–grassland junctions were unstable, the transformation of grassland into desert and converting desert to other unused land increased soil erosion by wind, apparently more than the opposite conversions. At the same time, the protection of deserts, other unused land, and grasslands by vegetation should be strengthened. With respect to erosion by water, the increase in grassland and woodland had a reducing effect, and the corresponding decrease from 2000 to 2018 was approximately $130.29 \times 10^4$ tons, occupying 52.17% of the total decrease in soil erosion by water owing to LUCC (Figure 9). In particular, when other unused land–grassland and woodland–grassland junctions were unstable, the conversion of other unused land to grassland and woodland to grassland reduced soil erosion by water substantially more than the opposite conversions. At the same time, changes in other unused land and grassland themselves

contributed to the reduction in water erosion, which was mainly driven by the increase in vegetation coverage. However, problems such as the conversion of cropland to other unused land or grassland increased the amount of soil erosion by water. This may be due to the fact that there was no vegetation protection at the early stage of transforming cropland back to grassland or abandoning cropland, which led to more severe soil erosion [60], and the vegetation restoration is slow in this circumstance.

The moduli of soil erosion by water and by wind differed depending on the vegetation type. The modulus of soil erosion by wind varied according to the following order: grassland > woodland > shrubbery, whereas the modulus of soil erosion by water differed as follows: woodland > shrubbery > grassland. For the former type of erosion, shrubbery and woodlands were more effective in intercepting wind and sand flow [59], but with respect to soil erosion by water, interception of rainwater by low grasslands and shrubbery likely had a larger effect [61]. Soil erosion by water decreased with an increase in vegetation coverage; however, the latter did not markedly reduce soil erosion by wind, probably because the increase in vegetation coverage in the Qaidam Basin has not yet reached 60%, which is considered the threshold vegetation coverage for reducing soil surface wind speed [5]. Forests, shrubs, and grasses contributed to the prevention of erosion by wind and water. Therefore, we recommend that an optimal arrangement of grasses, shrubs, and trees is needed at the junctions between other unused land and grassland or desert, and between desert and grassland, to prevent compound erosion by wind and water.

## 5. Conclusions

Climate change is helping with land cover improvement and thereby reducing wind erosion. But erosion by water has shown an increasing trend, which requires urgent protective measures against these types of soil erosion. In particular, strong protection against the compound erosion by wind and water is needed in areas such as Dulan.

Simultaneously, the rapid economic development prompted drastic changes in the type of land use in the Qaidam Basin. From 2000 to 2018, approximately 12.96% of the total area underwent land use type conversion, and such changes mainly included the expansion of cropland, woodland, and grassland at the expense of shrubbery, desert, and other unused land. Land use/cover changes are positive to the soil erosion of water but negative to the soil erosion of wind. Among them, increased vegetative cover and area of grasslands are beneficial in reducing soil erosion by wind and water. Our findings indicate that the increase in grasslands, shrubs, and trees in the Qaidam Basin had an overall positive impact on reducing soil erosion by wind and water.

Caution is required while interpreting the findings of this study. This study compared with other published results confirms that the RWEQ and RUSLE models applied can be used to assess soil erosion by wind and water in this region. However, Qaidam Basin extends over a vast territory with varying climate conditions. There are the limitations of an absence of field data of different regions in the Qaidam Basin for calibration and validation. Further long-term experimental and monitoring data from different regions in the Qaidam Basin and a careful revision of RWEQ and RUSLE parameters are required to ensure the accuracy of the models which could inform future research. In addition, there is a lack of availability of long-term and quality remote sensing data. The time period undertaken in the present study (2000–2018) is too short to generate more robust results. This aspect would also require further improvement in future work.

**Author Contributions:** Conceptualization, X.C.; methodology, Y.C.; software, L.B.; validation, J.L.; formal analysis, J.J. (Jinshi Jian); resources, X.M.; data curation, X.C.; writing—original draft preparation, X.C.; writing—review and editing, X.C.; visualization, Y.C.; supervision, J.J. (Juying Jiao); project administration, J.J. (Juying Jiao); funding acquisition, J.J. (Juying Jiao). All authors have read and agreed to the published version of the manuscript.

**Funding:** This study was supported by the second Tibetan Plateau Scientific Expedition and Research Program (STEP) (No. 2019QZKK0603) and the Strategic Priority Research Program of Chinese Academy of Sciences (No. XDA20040202).

**Data Availability Statement:** The data presented in this study are available on request from the corresponding author.

**Conflicts of Interest:** The authors declare no conflict of interest.

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
