# Peer review of "Impact of Land Use/Cover Changes on Soil Erosion by Wind and Water from 2000 to 2018 in the Qaidam Basin"

_land, doi:10.3390/land12101866_

Round 1
Reviewer 1 Report
I would like to thank the authors for their efforts in writing the paper. The paper is relatively well written and tries to mix up wind and water erosion as a result of Land use and land cover impacts. However, the paper should be enhanced according to the following comments:
- The abstract is good, but it needs to be adjusted slightly. Try to focus on the main results of the paper.
- The introduction is globally good, but some adjustments must be made: the originality of the paper as well as the scientific contribution are missing. The authors must add a whole section on the different applied models (based on the literature) for erosion simulation and quantification. The authors must focus especially on the two applied models for wind and water erosion (RWEQ and RUSLE).
- Figures are too small, illegible, and not clear at all. They must be enhanced and modified.
- The materials and ad data are too short and do not well cite the applied methodology, either for RUSLE or RWEQ: How is the data being processed? This section must be enhanced; consider adding a detailed flowchart for each model. The validation approach of the models is not cited at all. Both models are too delicate to process because they implement many factors, so the methodology must be accurately described.
- I did not distinguish in this section the originality of the paper; what is the scientific use of these applied models?
The authors are focusing on the simple use of models rather than the scientific contribution and the research side. A simple application approach does not bring any scientific soundness and makes the paper lose its originality, if it exists.
Results:
- temporal and spatial changes in LULC changes: how it was processed? Which classifier have you used? how it was operated? What is its accuracy? On what basis were they validated?
- The authors are discussing major conversion changes, but they must be based on a transition matrix. It should not be explained based on simple facts.
The validation approach is not discussed or introduced at all.
- The authors should rely on similar case studies to discuss the obtained results.
In general, the paper makes simple use of simulation models and lacks any originality. If any validation approach has been introduced, the results of the model must be extensively discussed and should be supported with literature references. I could not interpret or read any of the inserted figures. The resolution is too low and cannot be read. The paper should be considered for a workshop or presentation event.
I do not think the paper is suitable for publication.
Author Response
Response to Reviewers’ Comments
Dear reviewer 1:
Thank you for your letter and for the interest concerning our manuscript entitled “Impact of land use/cover changes on soil erosion by wind and water from 2000 to 2018 in the Qaidam Basin” (ID: Land-2528740) on Land. We also thank the reviewer for your great suggestions which improve the quality of this paper. We made one-to-one response to each comment rose by the reviewer and modified the manuscript accordingly.
Response to reviewer 1
We have revised manuscript according to the comments of the reviewer #1. Revised portion were marked as yellow color in the revision, changes marked manuscript.
Reviewer #1: I would like to thank the authors for their efforts in writing the paper. The paper is relatively well written and tries to mix up wind and water erosion as a result of Land use and land cover impacts. However, the paper should be enhanced according to the following comments:
Comment (1): The abstract is good, but it needs to be adjusted slightly. Try to focus on the main results of the paper.
Response (1): Thank you for your valuable advice. We have added the main results of the paper“Land use/cover changes is positive to soil erosion of water but negative to soil erosion of wind. In particular, the changes in vegetation coverage of other unused land and grassland contributed to 83.19% of the total reduction of soil erosion by water. Converting other unused land to grassland reduced by 94.69% of the total reductions in soil erosion by wind.” according to your suggestion. (see Line 29-32, Page 1)
Comment (2): The introduction is globally good, but some- adjustments must be made: the originality of the paper as well as the scientific contribution are missing. The authors must add a whole section on the different applied models (based on the literature) for erosion simulation and quantification. The authors must focus especially on the two applied models for wind and water erosion (RWEQ and RUSLE).
Response (2): Thank you for your valuable advice. We have added a whole section on the different applied models for erosion simulation and quantification in the introduction and cited relevant references (see Line 67-80, Page 2 and Line 590-620, Page 19-20).
Comment (3): Figures are too small, illegible, and not clear at all. They must be enhanced and modified.
Response (3): Thank you for your valuable advice. We have enhanced and modified the Fig. 2 to 11.
Comment (4): The materials and ad data are too short and do not well cite the applied methodology, either for RUSLE or RWEQ: How is the data being processed? This section must be enhanced; consider adding a detailed flowchart for each model. The validation approach of the models is not cited at all. Both models are too delicate to process because they implement many factors, so the methodology must be accurately described.
Response (4): Thank you for your valuable advice. We have enhanced the description of the methodology (see Line 142-205, Page 4-6). It's true that we've missed validation, so we rely on similar case studies to discuss the obtained results or compare with other published results to validate the results of the model. (see Line 389-400, Page 15)
Comment (5): I did not distinguish in this section the originality of the paper; what is the scientific use of these applied models? The authors are focusing on the simple use of models rather than the scientific contribution and the research side. A simple application approach does not bring any scientific soundness and makes the paper lose its originality, if it exists.
Response (5): In the methodology section, we are really not innovative enough. We aim to use modeling to explore the compound effects of land use/cover changes on soil erosion by wind and water.
Comment (6): temporal and spatial changes in LUCC changes: how it was processed? Which classifier have you used? how it was operated? What is its accuracy? On what basis were they validated?
Response (6): Our land-use data are based on the China Multi-Period Land Use Remote Sensing Monitoring Dataset (CNLUCC), which is a thematic database constructed by manual visual interpretation and validation using Landsat remote sensing images as the main source of information. The data are classified using the secondary classification system, and we use the first classification system, which is divided into six categories, namely, cropland, woodland, grassland, water, construction land, and unused land. Temporal and spatial variations in LUCC changes are obtained by overlaying the land use layers of the two years for 2000-2010, 2010-2018, and 2000-2018 in ArcGIS 10.7, respectively.
Comment (7): The authors are discussing major conversion changes, but they must be based on a transition matrix. It should not be explained based on simple facts. The validation approach is not discussed or introduced at all.
Response (7): Thank you very much. We are basing our discussion of the major conversion changes on the transfer matrix (e.g., Figures 3, 7, and 9). The discussion leading to these changes is mainly placed in Discussions 4.2 and 4.3 for illustration. We do lack validation only by discussing the results with other published results. (see Line 465-468, Page 17)
Comment (8): The authors should rely on similar case studies to discuss the obtained results.
Response (8): We completely agree with your viewpoints. In the discussion section, we add similar case studies to support the obtained results. (see Line 389-400, Page 15 and Line 465-468, Page 17)

Reviewer 2 Report
Evaluating the LUCC influence on soil erosion of wind and water is meaningful for improving the regional ecosystem services and sustainable development. This study used the Revised Wind Erosion Equation (RWEQ) and Revised Universal Soil Loss Equation (RUSLE) to verify changes in the extent of soil erosion by wind and water in the Qaidam Basin from 2000 to 2018 and the impact of LUCC on them. And then the authors tried to clarify the applications of these conclusions. The research and results are interesting. However, before this paper can be accepted for publication in the Land, the authors must appropriately deal with the following major and minor comments.
1). How reliable are the results of correlation analysis in the research? I noticed that the Land use changes were only included three years, however, the soil erosion effects of wind and water were only two stages? and how about the time matching.
2). According to your clarifications, the unused land areas accounted for the largest proportion in the three years. However, have you compared the soil erosion changes from wind and water in unused land and other unchanged land uses with the changed land uses.
3). Lines 125-127, did you present the original RUSLE from Renard et al. (1997), if not, clarify the revision source. As far as know, the RUSLE should be described by A = R × K × LS × C × P.
4). The Figures and attached labeling in the manuscript are all too small and unclear to be read, and I can’t get the initial intention to convey even though magnifying them.
5). This manuscript focused on the analysis of the impacts of LUCC. However, did the authors consider the significance and actual theoretical implications thoroughly. For example, in Lines 29-30, it need to be specific, and corresponding discussions about this recommendation need to be involved in the manuscript.
6). Lines 24-25, Please confirm these transformations being Positive or Negative to soil erosion of wind and water, did they all showed the reduction effects? and are different types of land uses the same?
7). Line 148, did the soil erosion grade be the Chinese Standard? And did the classifications need to be modulate in varying regions?
8). Lines 159-161, confirm the references of the mentioned software and apply the proper citing format.
9). Lines 145-146, there appeared Chinese characters in Table 2.
Author Response
Response to Reviewers’ Comments
Dear reviewer 2:
Thank you for your letter and for the interest concerning our manuscript entitled “Impact of land use/cover changes on soil erosion by wind and water from 2000 to 2018 in the Qaidam Basin” (ID: Land-2528740) on Land. We also thank the reviewer for your great suggestions which improve the quality of this paper. We made one-to-one response to each comment rose by the reviewer and modified the manuscript accordingly.
Response to reviewer 2
We have revised manuscript according to the comments of the reviewer #2. Revised portion were marked as yellow color in the revision, changes marked manuscript.
Reviewer #2: Evaluating the LUCC influence on soil erosion of wind and water is meaningful for improving the regional ecosystem services and sustainable development. This study used the Revised Wind Erosion Equation (RWEQ) and Revised Universal Soil Loss Equation (RUSLE) to verify changes in the extent of soil erosion by wind and water in the Qaidam Basin from 2000 to 2018 and the impact of LUCC on them. And then the authors tried to clarify the applications of these conclusions. The research and results are interesting. However, before this paper can be accepted for publication in the Land, the authors must appropriately deal with the following major and minor comments.
Comment (1): How reliable are the results of correlation analysis in the research? I noticed that the Land use changes were only included three years, however, the soil erosion effects of wind and water were only two stages? and how about the time matching.
Response (1): The effects of wind and water on soil erosion are analyzed in correspondence with land use changes. We calculated the modulus of soil erosion by wind for three years, but due to the length of the article, the change in soil erosion by wind and water from 2000-2018 was selected to be analyzed with the corresponding land use change from 200-2018.
Comment (2): According to your clarifications, the unused land areas accounted for the largest proportion in the three years. However, have you compared the soil erosion changes from wind and water in unused land and other unchanged land uses with the changed land uses.
Response (2): Thank you for your valuable advice. In Result section 3.3, we compare the soil erosion changes from wind and water in unused land and other unchanged land uses with the changed land uses. (see Line 329-375, Page 12-13)
Comment (3): Lines 125-127, did you present the original RUSLE from Renard et al. (1997), if not, clarify the revision source. As far as know, the RUSLE should be described by A = R × K × LS × C × P.
Response (3): We completely agree with your viewpoints that the RUSLE should be described by A = R × K × LS × C × P. The reason we multiply by a factor of 100 is to be consistent with the units of the mean modulus of soil erosion by wind. (see Line 184, Page 5)
Comment (4): The Figures and attached labeling in the manuscript are all too small and unclear to be read, and I can’t get the initial intention to convey even though magnifying them.
Response (4): Thank you for your valuable advice. We have enhanced and modified the Fig. 2 to 11.
Comment (5): This manuscript focused on the analysis of the impacts of LUCC. However, did the authors consider the significance and actual theoretical implications thoroughly. For example, in Lines 29-30, it need to be specific, and corresponding discussions about this recommendation need to be involved in the manuscript.
Response (5): Thanks for the great advice. We have discussed this recommendation in detail in Discussion 4.3 and provide corresponding evidence with other published results to support this recommendation. (see Line 478-497, Page 17)
Comment (6): Lines 24-25, Please confirm these transformations being Positive or Negative to soil erosion of wind and water, did they all showed the reduction effects? and are different types of land uses the same?
Response (6): Thank you for your valuable advice. These transformations did not all show the reduction effects. Our research has found that land use/cover changes resulted in an increase of 3.55×108 tons, a decrease of 1.85×108 tons, so land use/cover changes is positive to soil erosion of water but negative to soil erosion of wind. At the same time, the effects of different types of land use on wind and water erosion are not the same. (see Line 29-30, Page 1)
Comment (7): Line 148, did the soil erosion grade be the Chinese Standard? And did the classifications need to be modulate in varying regions?
Response (7): Yes. The soil erosion classes used in this paper are adopted from the Chinese standard. which is nationally standardized. It is true that the rate of loss varies from region to region, and the revision of this classifications standard by the Ministry of Water Resources is currently underway.
Comment (8): Lines 159-161, confirm the references of the mentioned software and apply the proper citing format.
Response (8): Thank you for your valuable advice. We have corrected it in the text. Change was made at Line 232, Page 7: “https://www.originlab.com” was replaced with “https://www.originlab.com, accessed on 1 March 2021”.
Comment (9): Lines 145-146, there appeared Chinese characters in Table 2.
Response (9): We are very sorry for our negligence. We have made corrections according to your comment. Changes were made in Table 2:“1:100 万” was replaced with “1:1 million”.

Reviewer 3 Report
The authors analyzed the changes of soil erosion by wind and water in the Qaidam Basin from 2000 to 2018, and the impact of LUCC on them. The work is interesting and helpful for the soil loss control and sustainable development in Qaidam. Some comments are as follows.
Wind and water erosion moduli were quantified, readers concern the reliability of the results. Are there validation of the results? Or at least could comparing with other published results.
Line 113, how was the Z value 50 determined?
In Line90-93, it is unnecessary to list the acreage of the study area in different provinces and which counties are included. Such information is not that interested by broad readers.
In Line105, 124, and 163, the models use the average climatic conditions to eliminate climatic effects. Please explain the specific method, were there averaged monthly values?
Line180-181, the authors introduced land use distribution by using toponyms, which was not marked in Fig.1 nor in Fig.2. It's reader-unfriendly. Same problem in Line195, 200, 212-213. Where are the Haixi Mongolian and Tibetan Autonomous Prefectures?
As a separate chapter of Discussion, 4.2 only introduces the changes of LUCC in detail without combining wind erosion and water erosion, which makes the article not coherent enough. Some of the content in 4.2 fits better in the Introduction.
The formatting of the manuscript needs to be adjusted, for example the formulas of Line 108-110 and Line 167-168, Table 2 and 4.
The quality of figures needs to be improved.
The language of the manuscript needs to be improved.
Author Response
Response to Reviewers’ Comments
Dear reviewer 3:
Thank you for your letter and for the interest concerning our manuscript entitled “Impact of land use/cover changes on soil erosion by wind and water from 2000 to 2018 in the Qaidam Basin” (ID: Land-2528740) on Land. We also thank the reviewer for your great suggestions which improve the quality of this paper. We made one-to-one response to each comment rose by the reviewer and modified the manuscript accordingly.
Response to reviewer 3
We have revised manuscript according to the comments of the reviewer #3. Revised portion were marked as yellow color in the revision, changes marked manuscript.
Reviewer #3: The authors analyzed the changes of soil erosion by wind and water in the Qaidam Basin from 2000 to 2018, and the impact of LUCC on them. The work is interesting and helpful for the soil loss control and sustainable development in Qaidam. Some comments are as follows.
Comment (1): Wind and water erosion moduli were quantified, readers concern the reliability of the results. Are there validation of the results? Or at least could comparing with other published results.
Response (1): We completely agree with your viewpoints. It's true that we've missed validation of the results. In the discussion section, we add similar case studies to support the obtained results. (see Line 389-400, Page 15 and Line 465-468, Page 17)
Comment (2): Line 113, how was the Z value 50 determined?
Response (2): Z is the maximum downwind erosion distance (m). In this study, we refer to Jiang et al.'s study in Qinghai Province to take the value 50.
Reference: Jiang, L.; Xiao, Y.; Ouyang, Z.Y.; Xu, W.H. and Zheng, H. Estimate of the wind erosion modules in Qinghai Province based on RWEQ model (in Chinese). Research of Soil and Water Conservation. 2015, 22, 21-25. https://doi.org/10.13869/j.cnki.rswc.2015.01.005.
Comment (3): In Line 90-93, it is unnecessary to list the acreage of the study area in different provinces and which counties are included. Such information is not that interested by broad readers.
Response (3): Thanks for the great advice. We have deleted “mainly including Delingha City, Golmud City, Dulan County, Haixi Mongolian and Tibetan Autonomous Prefecture, Ulan County, Maduo County, Zhiduo County, and Qumalai County,” according to your suggestion. (see Line 111-113, Page 3)
Comment (4): In Line105, 124, and 163, the models use the average climatic conditions to eliminate climatic effects. Please explain the specific method, were there averaged monthly values?
Response (4): It is really true as you think that the specific method uses multi-year averaged monthly values. We have modified it. Change was made at Line126, 173, and 234, Page 3, 5, 7: “multi-year average climatic conditions” was replaced with “multi-year averaged monthly values”.
Comment (5): Line180-181, the authors introduced land use distribution by using toponyms, which was not marked in Fig.1 nor in Fig.2. It's reader-unfriendly. Same problem in Line195, 200, 212-213. Where are the Haixi Mongolian and Tibetan Autonomous Prefectures?
Response (5): Special thanks for your good comments. Considering your suggestion, we have added toponyms in Fig.1.
Comment (6): As a separate chapter of Discussion, 4.2 only introduces the changes of LUCC in detail without combining wind erosion and water erosion, which makes the article not coherent enough. Some of the content in 4.2 fits better in the Introduction.
Response (6): Thank you very much. But discussion 4.2 introduces the changes of LUCC and its influencing factors, in order to set the stage for Discussion 4.3 the impact of LUCC on erosion by wind and water.
Comment (7): The formatting of the manuscript needs to be adjusted, for example the formulas of Line 108-110 and Line 167-168, Table 2 and 4.
Response (7): Thank you for your careful work. We have carefully revised the formulas of Line 130-132 and Line 239-240, Table 2 and 4 in the text.
Comment (8): The quality of figures needs to be improved.
Response (8): Thank you for your valuable advice. We have enhanced and modified the Fig. 2 to 11.

Reviewer 4 Report
It is a good study to see the effect of land uses vis-a-vis erosion pattern particularly in view of climate change. I congratulate you for a well prepared manuscript. Please look into the following observations:
I found the conclusion a bit similar to discussion. One may not call for urgent protection measures if climate change influencing on land use change was found to have some positive effect. It was said that climate change is helping land cover improvement and thereby reducing wind erosion. I feel more clarity in conclusion is desired.
Please change the Fig. 2 to 11 as all the figures have very poor clarity and legends are unreadable.
Do you find any importance or relevance of Fig.12. I feel it is redundant and not giving any important information.
Line 324: Please change the format of citation.
If not very important, you may drop your own citation.
Author Response
Response to Reviewers’ Comments
Dear reviewer 4:
Thank you for your letter and for the interest concerning our manuscript entitled “Impact of land use/cover changes on soil erosion by wind and water from 2000 to 2018 in the Qaidam Basin” (ID: Land-2528740) on Land. We also thank the reviewer for your great suggestions which improve the quality of this paper. We made one-to-one response to each comment rose by the reviewer and modified the manuscript accordingly.
Response to reviewer 4
We have revised manuscript according to the comments of the reviewer #4. Revised portion were marked as yellow color in the revision, changes marked manuscript.
Reviewer #4: It is a good study to see the effect of land uses vis-a-vis erosion pattern particularly in view of climate change. I congratulate you for a well prepared manuscript. Please look into the following observations:
Comment (1): I found the conclusion a bit similar to discussion. One may not call for urgent protection measures if climate change influencing on land use change was found to have some positive effect. It was said that climate change is helping land cover improvement and thereby reducing wind erosion. I feel more clarity in conclusion is desired.
Response (1): Thanks for the great comment. We have made corrections according to your suggestion. Changes were made at Line 502-506, Page 17: “erosion by wind in the Qaidam Basin continued to decrease from 2000 to 2018.” was replaced with “climate change is helping land cover improvement and thereby reducing wind erosion”.
Comment (2): Please change the Fig. 2 to 11 as all the figures have very poor clarity and legends are unreadable.
Response (2): Thank you for your valuable advice. We have enhanced and modified the Fig. 2 to 11.
Comment (3): Do you find any importance or relevance of Fig.12. I feel it is redundant and not giving any important information.
Response (3): Thanks for the great advice. Change was made at Line 451-452, Page 16: “Fig.12.” was deleted.
Comment (4): Line 324: Please change the format of citation.
Response (4): We are very sorry for our incorrect writing. The right format is “Teng et al. [45]”. (see Line 410-412, Page 15)
Comment (5): If not very important, you may drop your own citation.
Response (5): Thank you for your valuable advice. Change was made at Line 714-716, Page 22: my own citation was deleted.

Round 2
Reviewer 2 Report
I noticed that Comment (1) I have valued before, it seems that you did not handle that problem properly. You have reply this problem with the reseaon of the manuscript length, however, that was not your excuse for this issue. Relevant results, such as the erosion calculated results, can be described in the text form, but that should be analysed.
Comment (7), you cited Equation (13) by Renard et al., however, you say it a Chinese standard?
There also have a few minor errors in the manuscript. For example, 1). Line 470, the Reference number 614? 2). Line 229 was disorderly described. 3). Lines 184, 185, the Equation presences, et al.
Author Response
Response to Reviewers’ Comments
Dear reviewer 2:
Thank you again for your letter and for the interest concerning our manuscript entitled “Impact of land use/cover changes on soil erosion by wind and water from 2000 to 2018 in the Qaidam Basin” (ID: Land-2528740) on Land. We also thank the reviewer for your great suggestions which improve the quality of this paper. We made one-to-one response to each comment rose by the reviewer and modified the manuscript accordingly.
Response to reviewer 2
We have revised manuscript according to the comments of the reviewer #2. Revised portion were marked as yellow color in the revision, changes marked manuscript.
Reviewer #2:
Comment (1): I noticed that Comment (1) I have valued before“How reliable are the results of correlation analysis in the research? I noticed that the Land use changes were only included three years, however, the soil erosion effects of wind and water were only two stages? and how about the time matching.”, it seems that you did not handle that problem properly. You have replyed this problem with the reason of the manuscript length, however, that was not your excuse for this issue. Relevant results, such as the erosion calculated results, can be described in the text form, but that should be analysed.
Response (1): We're sorry we didn't handle this problem properly last time. We completely agree with your viewpoints. It's true that we've missed validation of the results. We could only confirm the obtained results by comparing them with the findings of Wang, Jiang, Teng, etc. in the discussion section 4.1. These findings indicate that our calculated modulus of soil erosion by wind and water were reliable. (see Line 378-385, 388-390, Page 14 and Line 400-402, 409-410, Page 15)
We calculated Land use changes and the modulus of soil erosion by wind and water for the years 2000, 2010, and 2018. The effects of wind and water on soil erosion are analyzed in correspondence with land use changes. The study found that the modulus of the average erosion by wind in 2000, 2010, and 2018 was 31,183.70, 24,705.20 and 23,547.60 [t/(km2·a)], with an average of 26,478.83 [t/(km2·a)], showing a continuous decreasing trend. And it found that the modulus of the average erosion by water in 2000, 2010, and 2018 was 692.97, 1,203.38 and 1,442.47 [t/(km2·a)], with an average of 1,112.94 [t/(km2·a)], showing a continuous increasing trend. Thus, the change in soil erosion by wind and water from 2000 to 2018 was selected to be analyzed with the corresponding land use change from 2000 to 2018. (see Line 375-378, 385-388, Page 14)
Comment (2): Comment (7)“Line 148, did the soil erosion grade be the Chinese Standard? And did the classifications need to be modulate in varying regions?”, you cited Equation (13) by Renard et al., however, you say it a Chinese standard?
Response (2): Thank you for your valuable advice. In this study, the modulus of the soil erosion by water was calculated by quoting the formula of Renard et al. And the grades of the modulus of the soil erosion by water were expressed according to the soil erosion grade standard (SL 190-2008), as shown in Table 3, which is the Chinese standard. Currently, the standard is nationally harmonized, but it is still being further developed. (see Line 205-208, Page 6)
Comment (3): There also have a few minor errors in the manuscript. For example, 1). Line 470, the Reference number 614? 2). Line 229 was disorderly described. 3). Lines 184, 185, the Equation presences, et al.
Response (3): Thank you for your careful work. We have carefully revised the errors in the text. We have made corrections according to your comment. Changes were made at Lines 58, 71, 150, 184-190, 229, 331, 379, 470: “planting trees and grass … reduces … intercepts … decreases…” was replaced with“planting trees and grasses … reduce … intercept … decrease…”; “CSLE” was replaced with “(CSLE)”; “The EF and SCF is” was replaced with “The EF and SCF are”; “ ” was replaced with “ ”; “the net change in the soil wind erosion modulus” was replaced with “the net change in the mean modulus of soil erosion by wind”; “Fig. 8” was replaced with “Figure 8”; “[48]” was replaced with “[49]”; “[614]” was replaced with “[61]”.
